# Evolution and Reassortment of H6 Subtype Avian Influenza Viruses

**DOI:** 10.3390/v15071547

**Published:** 2023-07-13

**Authors:** Mingqin Lin, Qiu-Cheng Yao, Jing Liu, Miaotong Huo, Yan Zhou, Minyi Chen, Yuanguo Li, Yuwei Gao, Ye Ge

**Affiliations:** 1College of Coastal Agricultural Sciences, Guangdong Ocean University, Zhanjiang 524088, Chinayqc198292@163.com (Q.-C.Y.);; 2Military Veterinary Research Institute of Academy of Military Medical Sciences, Changchun 130000, Chinayuwei0901@outlook.com (Y.G.)

**Keywords:** avian influenza virus, H6 subtype, genetics and evolution, reassortment, prevention and control

## Abstract

The H6 subtype of avian influenza virus (H6 AIV) is the most detected AIV subtype in poultry and wild birds. It causes economic losses to the poultry industry, and the most important, H6 AIV may have the ability to infect mammals, which is a great threat to public health security. In addition, the H6 subtype can serve as a precursor to providing internal genes for other highly pathogenic AIVs, posing a potential threat. H6 AIV currently face to the high positive detection rate and harmless nature of H6 AIV and because not highly effective H6 subtype vaccine available on the market. In this study, we focused on the prevalence of H6 AIV in poultry and wild birds, phylogenetic analysis, genetic variation characteristics, selection analysis, and prevention and control to provide relevant references for the scientific prevention and control of H6 AIV in future.

## 1. Introduction

Avian influenza viruses are influenza A (type A) viruses, which are human-animal pathogens. Highly pathogenic avian influenza viruses (HPAIVs) and low pathogenic avian influenza viruses (LPAIVs) can be classified according to their pathogenicity for chickens [1]. Influenza A virus consists of 8 independent RNA fragments (PB2, PB1, PA, HA, NP, NA, M, and NS, ordered from large to small), which can be divided into 18 HA subtypes (H1-H18) and 11 NA subtypes (N1-N11) according to the difference in hemagglutinin (HA) between neuraminidase (NA) [2,3].

AIV has a wide range of hosts, and it is now generally accepted that shorebirds and waterfowls are the natural reservoir (except for H17N10 and H18N11 [4]). Wild birds played the important role in the evolution and spread of avian influenza viruses [5]. Different avian influenza subtypes have been isolated in domestic birds, including some subtypes such as H5 and H7, which have caused the death of many domestic poultry and wild birds. H5, H6, H7, H9 and H10 subtypes of avian influenza viruses even have caused human infection [6,7,8,9].

In 1965, the H6 subtype was first isolated from turkeys in Massachusetts, USA [10] However, according to the sequences information from GISIAD database, the earliest H6 sequences were found in Canada: A/Turkey/Canada/1963(H6N8) and A/Turkey/Canada/1963(H6N2). The H6 subtype of AIV can infect the largest number of species [11], such as wild birds, poultry, and mammals. Moreover, positive detection rates for H6 AIV are increasing in both wild and domestic waterfowl and terrestrial birds [12]. Today, all H6 AIVs found have been LPAIV [13]. Despite H6 AIV being an LPAIV, A/teal/HK/W312/97 (H6N1) has been identified as a potential precursor of A/Hong Kong/156/1997 (H5N1) [14], and some of the H6NX strains have been identified as donors for H5 and H9N2 subtype AIV [15,16]. Such reassortments may pose a major threat to public health. Studies have reported that the isolated H6 virus subtype can infect mammals such as mice and ferrets without being adapted to the host [14,17]. Additionally, Xu et al. found that some H6N6 viruses isolated in China can cause serious illness and even death in mice [18]. Infection with H6 AIV has also been detected in larger mammals. For example, H6N6 was detected from pigs in eastern and southern China in 2009 and 2010 (A/swine/Yangzhou/080/2009(H6N6) and A/swine/Guangdong/k6/2010(H6N6)), respectively [19,20]. Besides, a dog infected with H6 AIV was detected in samples from Taiwan (A/canine/Taiwan/E01/2014(H6N1)) [21]. Most importantly, the first human case of H6N1 AIV infection was reported in Taiwan, China, in 2013 [8].

In this study, we downloaded the H6 avian influenza virus sequences from the GISIAD and NCBI databases and removed the repeat sequences. Then, we performed statistical analysis on sequence information such as separation time, isolation location, species, etc., and drew a structural simulation figure of H6-HA protein to analyze the genetic evolution and biological characteristics of the H6 subtype of avian influenza strains isolated thus far to illustrate the current evolutionary characteristics of this virus subtype in relation to its preventive control.

## 2. Epidemiological Characteristics of Avian Influenza Virus H6 Subtype

### 2.1. The Geography Distribution of H6 Subtype

There was a total of 3234 HA sequences for the H6 subtype of AIV, 638 NA sequences of the N1 subtype of AIV, 1069 N2 sequences, 17 N3 sequences, 16 N4 sequences, 104 N5 sequences, 801 N6 sequences, 9 N7 sequences, 308 N8 sequences, and 19 N9 sequences until Jan 2023.

A foreign surveillance study including more than 36,000 wild birds from Europe and America (1998–2006) found that H6 was the most frequently detected influenza virus subtype [22]. H6NX AIV spread from North America to Eurasia around 1976, with a population size increase around 1990, peaking around 2015, declining after 2015, and maintaining stable levels after 2018 [13]. As of 3 October 2022, GISAID included a total of 3234 H6 AIV sequences (HA), of which H6N2 had the highest detection rate (1069 strains) and H6N7 had the lowest number (9 strains), and the number of each subtype was shown in Figure 1.

There were nine NA subtypes of H6NX, with North America, Asia, and Europe having the largest distribution in terms of number and subtype variety. The main distribution of H6NX subtypes were as follows: North America and Asia covered nine subtypes, South America and Africa were dominated by H6N1, H6N2, and H6N8, Europe was dominated by H6N1, H6N2, H6N5, H6N6, and H6N8, and Oceania included H6N1, H6N2, H6N5, and H6N9 [11], and notably, a strain of H6N8 AIV was isolated in Antarctica in 2011. The global distribution of H6NX is shown in Figure 2a–g.

China, the epicenter of influenza [23], has an extremely high isolation rate of H6 subtypes [13]. N1, N2, and N6 were the major H6 AIV subtypes in China, with duck derived H6N2 predominating from 2000–2005. After that, N2 and N6 replaced N1 subtypes as the main NA subtypes of H6-subtype AIVs in China, and the proportion of H6N6 subtypes has gradually increased since 2009 [13,16]. In China, the H6 subtype AIV is mainly concentrated in the southern region, with Guangdong Province having the largest number of isolated strains of 608 strains, followed by Hunan Province and Fujian Province. Hong Kong has the richest NA subtypes, with 7 NA subtype sequences isolated except N3 and N6, followed by Hunan Province and Jiangxi Province, with 6 different NA subtype strains (Figure 3). The global epidemiological trend of H6NX subtypes from 2001–2021 (Figure 4) showed that the H6N2 subtype was the dominant subtype from 2001 to 2005, with the N6 subtype becoming increasingly prevalent after 2013 and the N1 subtype decreasing overall after 2008. This trend is essentially in line with the trend of H6 AIV in China.

### 2.2. The Species Distribution of H6 Subtype

Although the H6 subtype of AIV exhibits low pathogenicity, it should be given some attention based on its broad host range (including wild birds, poultry, pigs, and humans) [11], internal genetic homology with highly pathogenic AIVs and high frequency of detection of multiple key amino acid mutations.

H6NX subtype AIV has been isolated from a variety of wild birds in many countries and regions, with a total of 1173 sequences (HA), and the isolation sources mainly included Anseriformes, shorebirds, etc., of which Anseriformes were the main hosts (79.96%, Figure 5). Monitoring revealed that the H6 AIV was detected throughout the year, but the highest isolation rate was observed in spring and winter [24].

China is located on the important migration routes of East Asia-Australia, Central Asia-India, and West Asia-Central Africa, and many migratory birds come here every year to spend the winter. These migratory birds may lead to transmissions of various AIVs in China. H6NX became the main prevalent subtype of AIV in wild birds in southeast China from 2015–2019 [25]. The eight representative H6 strains isolated from wild birds in eastern China and the H6 sequences detected in Poyang Lake, China, indicate that as birds migrate, AIVs undergo not only intercontinental but also intracontinental gene exchange, including gene flow between the same subtype of AIV and different subtypes (including the highly pathogenic H5 subtype), reflecting the genetic evolutionary diversity of H6NX AIV from wild birds in China and the potential threat of cross-species transmission [15,17].

In addition, genetic reassortment between wild birds and domestic poultry is relatively common. We created this phylogenetic tree (Figure 6) by screening the H6 AIV (HA) sequences by country and species. The result showed that H6-HA had evolved into two mainly branches. Both branches both had wild birds and poultry clustered, suggesting that there may have been a reassortment of the H6-HA gene between them. Although this lineage is mainly dominated by wild bird source sequences, they do not all form a small branch independently but are clustered together with more duck derived sequences and some chicken derived sequence. It indicates that the reassortment of H6 AIV may have occurred in two modes: wild bird-aquatic domestic (duck) and wild bird-terrestrial domestic (chicken). In addition, it has been noted that there are some sequences isolated from wild birds in North American countries in the Eurasian lineage, so it can be speculated that H6 AIV appears to have spread intercontinental with the migration of wild birds. Researchers have also reported on these phenomena. For example, all genomes of the avian influenza strain A/wild bird/Hubei/01.24_FHC172-1/2017 (H6N6) isolated from central China were from poultry, suggesting that the H6 AIV is spilling over from poultry to wild birds [25]. Moreover, Ge et al. showed a variety of recombination phenomena from poultry and wild bird-derived viruses in the 13 wild bird derived H6 strains isolated in China [26]. The above examples suggest and phylogenetic analysis that such reassortment may not be uncommon, suggesting that we should strengthen the monitoring of wild birds.

Among poultry species, researchers have concluded through long-term monitoring that ducks are more likely to be the primary host of H6 AIV than geese and chickens [11]. A total of 1237 duck originated H6 sequences (HA) have been published, accounting for the largest proportion of poultry (1237/1934, Figure 7). H6 AIV distribution in wild birds. The numbers of H6 virus isolates in different wild birds are shown according to their submission information in GISAID. H6N2 was predominantly prevalent in domestic ducks from 2000–2005, with H6N6 gradually replacing H6N2 after 2006 and becoming the predominant prevalent subtype during 2011–2020 (Figure 8). Waterfowl are known to be natural hosts for AIV, and domestic ducks, as an important component, play a role in the evolution of AIV transmission and recombination. Although early literature indicated that H6 AIV has barriers to interspecific transmission, transmission between domestic ducks and terrestrial poultry is uncommon [27]. However, the recent BSSVS procedure by Yuan Z et al. showed that ducks were the main source for introducing H6 into chickens and geese [16].

Chickens and geese were also the main poultry infected with H6 AIV, accounting for 16% and 12%, respectively. Most subtypes were isolated from them (Figure 9a,b). Chickens infected with H6 AIV have exhibited aggravated pathological changes in recent years. In addition, researchers found that chicken colonic cells can express avian and human-like receptors, presumably acting as intermediate hosts for the transmission of influenza viruses to humans [28].

Additionally, H6 AIV has been isolated from mammals and humans. Avian-derived H6N6 AIV was isolated from domestic pigs in Yangzhou and Guangdong in 2009 and 2010, respectively. The first human case of H6 subtype AIV infection was reported in Taiwan, China, in 2013. Additionally, the viruses can replicate efficiently in experimental animals without adaptation.

## 3. Phylogenetic Analysis of H6Nx Avian Influenza Virus

Phylogenetic analysis of the AIV genes in this study was performed to assess their relationships with corresponding genes in domestic poultry, wild birds and other animals. The H6 subtype of avian influenza viruses is distributed on all continents. Based on geographic location and phylogeny, it can be divided into North American (NAm) and Eurasian (EA) lineages, with additional studies suggesting the existence of a South American influenza virus lineage (Sam lineage) [2,29,30]. The Eurasian genealogy is predominant and further divided into ST339-like, ST2853-like and HN57-like subbranches [27]. The ST339-like and ST2853-like branches of H6Nx are the two main prevalent branches in China [16].

Since the H6 subtype was first reported, reassortment with other influenza A viruses has occurred, leading to diversification of the viral genome. Cui J et al. performed phylogenetic analyses on H6 AIVs in Chinese farms from 2014–2018 and showed that 19 different genotypes were formed among 20 representative H6 subtype strains and that the internal genes of these viruses showed complex associations with different subtypes of AIV across continents [31]. Genetic recombination in the North American (Nam)-Eurasian (EA) lineage: for geographical reasons, the Nam and EA lineages do not generally exchange in poultry, but influenza viruses from these two lineages often reassorted in wild birds [32]. Early studies also indicate that there have been multiple invasions of Eurasian H6 avian influenza viruses into North American [33]. In addition, there is evidence of multiple reassortments or genetic exchanges between migratory waterfowl and poultry [14,25,34], so wild bird-poultry interactions may potentially facilitate the evolution and birth of new genotypes. H6 AIV (LPAIV)-HPAIV/LPAIV rematch: A/teal/HK/W312/97(H6N1) is a potential precursor of HK/97 H5N1 virus, and the remaining seven gene fragments except HA are highly homologous to HK/97 H5N1 [14]. Genetic evolutionary analysis of the first infected human A/Taiwan/2/2013 (H6N1) indicates that three internal genes of the virus originated from H5N2 (PB2, PA, M) [35]. In 2021, MingXian Gui et al. isolated a recombinant H6N6 AIV strain from chickens with PB2, PA from H5N2, and NA from H5N6 [13]. The above studies suggest that the H6 subtype of AIV is continuously exchanging genes with HPAIV (H5NX). Some of the H6N2 subtype AIV isolated in South China in 2014 were classified as the H9N2 gene pool, indicating frequent rearrangements of H6N2 and H9N2 viruses in aquatic avian-derived influenza viruses [16]. Chuan Xia Hu found two H6N8 strains closely related to the early Mongolian duck derived H3N8 virus, and one strain closely related to the NA gene of the 2014 Japanese epidemic duck H3N8 in avian influenza surveillance in wild birds in eastern China from 2016–2017 [17].

Additionally, H6NX has been shown to act as a gene pool for highly pathogenic AIV with internal gene exchange with HPAIV, facilitating the occurrence of reassortment events and the formation of potent virulent genotypes. Although only one incident of H6N1 AIV infection in humans has been identified and did not result in patients with severe clinical signs, H6 AIV has been isolated from pigs, and many H6N2 strains have been found to prefer the human receptor [36]. This suggests that H6 AIVs may be undergoing receptor-adaptive mutations, which increases the likelihood of a pandemic in the population. H6 AIV facilitates the emergence and global spread of novel AIVs through wild bird-poultry interactions that promote reassortment across the Eurasian-North American spectrum and between genes of different subtypes of AIVs. Given the importance of wild waterfowl and poultry in the evolution and spread of the H6 AIV, there is a need to strengthen ongoing surveillance of wild birds and poultry, especially wild waterfowl, ducks and chickens, and some attention needs to be paid to domestic geese as well.

## 4. Molecular Characterization of Surface Genes and Inner Genes

Although H6 AIV acts as LPAIV, its six internal genes have been found to continuously rearrange with other AIV subtypes (LPAIV/HPAIV) to acquire new genotypes since surveillance [16,17], with mutations in receptor adaptation, drug resistance, and pathogenicity, posing a significant risk to public health. 

Researchers evaluated a series of H6 viruses isolated from live poultry markets in southern China and found that 34% of the viruses were able to bind human receptors [12], suggesting that they may break the interspecies barrier and undergo cross-species transmission. AIV binds mainly to the sialic acid α2-3 galactosidase (Siaα2-3Gal) receptor (avian receptor) and acquires recognition of the sialic acid α2-6 galactosidase (Siaα2-6Gal) receptor (human receptor), so the ability of the AIV to infect humans depends on the ability to specifically recognize the Siaα2-6Gal receptor.

Studies show that the presence of amino acid mutations in HA and PB2 is strongly associated with the ability of AIVs to infect humans [37]. HA mutations: Findings point to Q226L in the HA protein as the key amino acid that increases the binding capacity of human-type receptors. Notably, a single amino acid mutation, Q226L, in HA enables H6N2 virus transmission in guinea pigs via respiratory droplets [38]. Mutagenesis work shows that E190V and G228S substitutions in HA are important for obtaining human receptor binding capacity and that P186L substitution reduces binding to the avian influenza receptor [39]. The trinity of N137, V190 and S228 in TW H6 (A/Taiwan/2/2013(H6N1)) HA may reduce the need for hydrophobic residues at HA_1_226 of H2 and H3 HAs when binding to human-like receptors, which makes H6 AIV being susceptible to infecting humans probably [40]; PB2 mutations: Discovery of the K702R mutation in the PB2 protein of H6N2 virus in South African chickens indicated that K702R likely plays a role in adaptation to mammals [37]. Meanwhile, a study found that PB2-E627K enhanced the replication of the H6N1 virus in mammals [41]. Other mutations: Kaliannan Durairal et al. identified PB1 (S375N/T) and PA (A404S, S409N) human host markers in the genes of KNU2019-48 (H6N6) isolates [42], which may contribute to the adaptation of H6 subtype AIVs to mammals. Furthermore, although amino acid mutations that have been previously shown to be associated with receptor preferences for other AIV subtypes have been identified in H6 AIV, where a small number of results have been based only on currently published AIV human host markers and have not been validated in animal studies, so further investigation is needed to clarify the relationship between these mutations and H6 AIV human receptor preferences.

Also, some amino acid mutations in internal genes were observed in H6 AIV. Zhi et al. generated H6N1 variants (produced by independent serial passage of A/Mallard/Sanjiang/275/2007 in mice, named MA-P8M1 and MA-P8M3) and they and wild-type strains were used separately for mouse infection experiments. They found that adaptations containing the PB2-E627K, PA-T97I, and HA-N394T mutations were highly pathogenic, indicating that these three mutations could increase the replication capacity and virulence of H6N1 AIV in mammals [43]. Experimental results by KaiHui Cheng et al. also confirmed that PB2-E627K and PA-T97I mutations enhance the ability of H6N1 viruses to replicate and cause disease in mammals [41]. However, the PB2-E627K mutation may not be the determining factor, as one study revealed a novel mechanism that may affect AIV virulence: the combination of HA (H156N, S263R) and PA (I38M) compensated for the lack of PB2 (627K) and enhanced the virulence of influenza A H6N6 virus adapted to mice [44]. In addition, it has been repeatedly reported that H6 AIVs can replicate efficiently in mouse lungs without adaptation, suggesting that some changes may be occurring in the pathogenicity genes of H6 subtypes of AIVs, yet the paucity of relevant studies in recent years has led to a failure in further understanding whether there are other potentially relevant amino acid mutation sites for H6 AIV.

Currently, therapeutic agents for avian influenza contain mainly M2 blockers as well as NA inhibitors. However, in recent years, M2-S31N and M2-V27I mutations have been found in some strains that are resistant to amantadine [25,45]. Mutant residues E119V, H275Y, R293K and N295S were found in the H6 AIV NA origin gene of Korean *Anas platyrhynchos*, indicating resistance to oseltamivir [42]. However, several H6 subtype AIV isolates have also been found to be susceptible to oseltamivir [46], so enhanced monitoring of the resistance of H6 subtype AIV against NA inhibitor drugs is needed to precisely select targeted antiviral drugs for efficient treatment. In addition, during avian H6 AIV surveillance in China, GX1167 (H6N6) was found to have a D92E mutation in the NS1 protein, which was able to increase the resistance of the virus to interferon [24].

## 5. Other Characters

### 5.1. Selection Analysis

Higher dN/dS ratios reflect both adaptive pressure and relaxed selective constraint that might facilitate the accumulation of favorable genetic changes associated with adaptive fitness to a new host population. Rates of nonsynonymous and synonymous substitutions (dN/dS) for each segment from H6NX AIV were estimated using Launch DnaSP6 and MEGA 7.0 software. Specifically, for each alignment, the sequences were selected without any annexed base. The sequences had the whole coding region.

Rates of nonsynonymous and synonymous substitutions (dN/dS) for each segment from H6NX AIV were estimated using the Launch DnaSP6 software and MEGA 7. The results showed that the HA, PB1 and NS segments experienced greater levels of selection pressure than the other five segments (Table 1 and Figure 10), which had significantly higher dN/dS substitution ratios.

### 5.2. The HA Protein Structure of H6 AIV

PyMOL was used to generate the model image. We selected A/chicken/Taiwan/0705/1999(H6N1) and A/Taiwan/2/2013(H6N1) to plot structure prediction. The HA protein structure of H6 AIV was built by homology modeling, and a suitable template was selected by using Alphafold2 for homology modeling (Figure 11).

## 6. Prevent and Control of H6 AIV

Vaccination is an important preventive and control measure to prevent viral infection and deters the spread of the virus. However, a H6 AIV vaccine is currently not commercially available. Nonetheless, progress has been made in H6 AIV cold-adapted vaccines [47], recombinant baculovirus vaccines [48,49], and plant-produced virus-like particle vaccines [50], which could be used as a basis for developing highly effective vaccines against H6 AIVs for scientific prevention in the future.

In summary, it is of great significance to pay attention to the recombination of H6 AIV internal genes and other subtypes of AIV genes. Therefore, we should determine the mutations of viral proteins that affect the pathogenicity of the H6 AIV and the adaptability of human receptors, investigate the pathways of drug resistance of H6 AIV, strengthen the monitoring of wild water birds and poultry, and develop specific drugs and preventive vaccines that are both safe and effective against H6 subtype AIV.

## 7. Conclusions

Avian influenza of the H6 subtype has been in a state of evolution and recombination. Although only one case of human infection has been reported as of today, there is a large body of research showing its ability to infect mammals, and there is evidence of its recombination with other subtypes of avian influenza, including the H5 subtype, which undoubtedly poses a public health threat. Therefore, we should continuously monitor its evolution and recombination, and develop highly effective vaccines to safely and effectively prevent and control H6 subtype avian influenza.

## Figures and Tables

**Figure 1 viruses-15-01547-f001:**
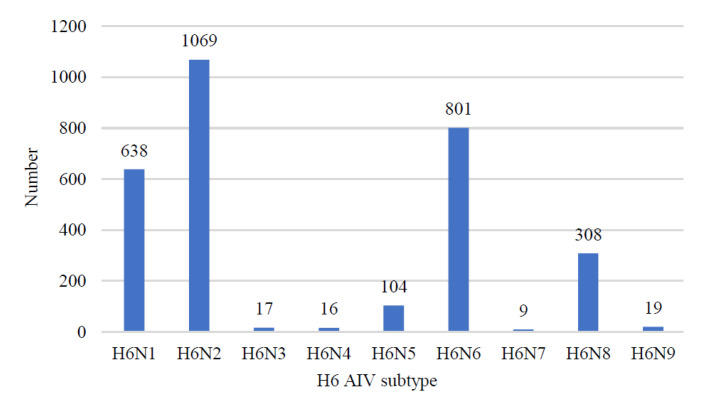
Statistical chart for the number of each subtype of H6 AIV. The HA sequences were downloaded from the GISAID databases, and 253 H6N0 (no typing NA) sequences were not included in this chart.

**Figure 2 viruses-15-01547-f002:**
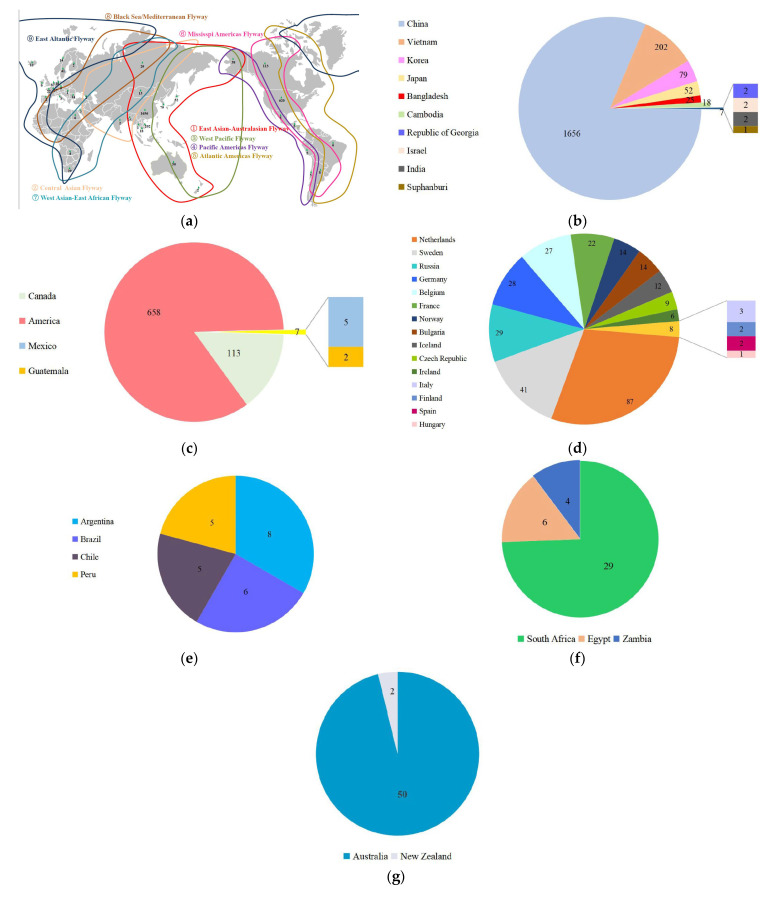
Global circulation of H6 viruses. (**a**) Global circulation of H6 viruses. The HA sequences were downloaded from the GISAID database, and then the isolation locations were noted on the map with green triangles (one H6 sequence in Antarctica was not labeled). (**b**) Asian circulation of H6 virus. (**c**) North American circulation of H6 virus. (**d**) European circulation of H6 virus. (**e**) South American circulation of H6 virus. (**f**) African circulation of H6 virus(**g**) Oceania circulation of H6 virus.

**Figure 3 viruses-15-01547-f003:**
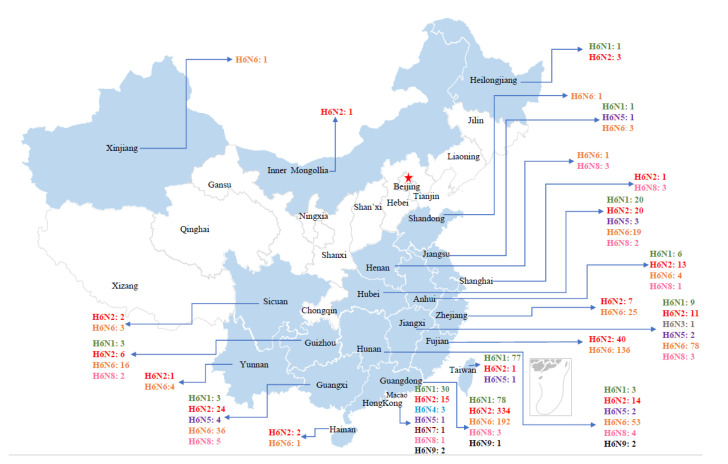
H6 AIV distribution in Chinese provinces. The HA sequences were downloaded from the GISAID databases. Provinces where H6 AIV was isolated are marked in light blue, and provinces where H6 AIV was isolated are marked in light gray. H6N1 is marked in green; H6N2 is marked in red; H6N3 is marked in gray; H6N4 is marked in blue; H6N5 is marked in purple; H6H6 is marked in orange; H6N7 is marked in brown; H6N8 is marked in pink; H6N9 is marked in black. There are also 201 H6N0 sequences and 101 sequences labeled only for China or East China that are not marked in this figure. The red star represents Beijing, the capital of China.

**Figure 4 viruses-15-01547-f004:**
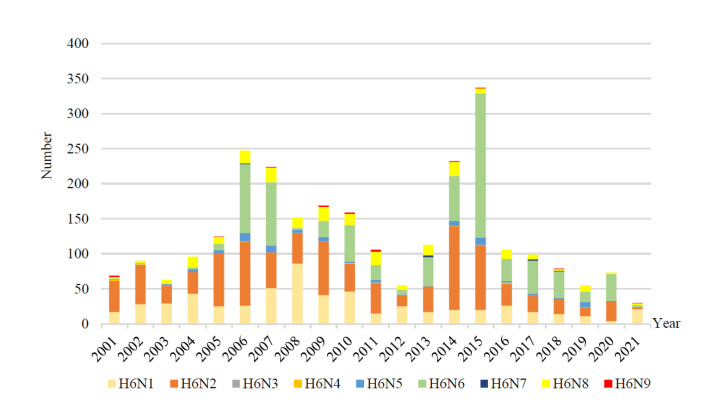
Statistical chart of the number of H6 AIV subtypes from 2001 to 2021. The HA sequences were downloaded from the GISAID databases, and H6N0 sequences were not included in this chart.

**Figure 5 viruses-15-01547-f005:**
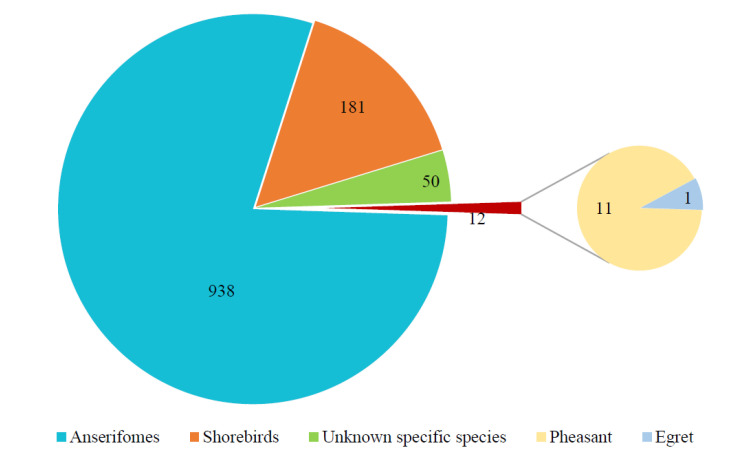
H6 AIV distribution in wild birds. The numbers of H6 virus isolates in different wild birds are shown according to their submission information in GISAID.

**Figure 6 viruses-15-01547-f006:**
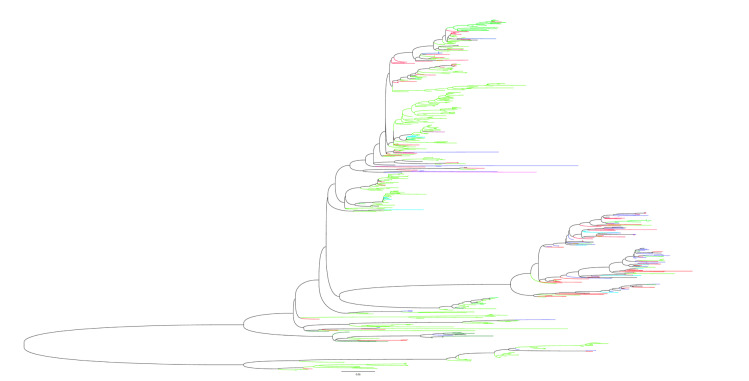
Phylogenetic analysis of H6 avian influenza viruses (HA). We select sequences according to different regions and species and then build an evolutionary tree. In this figure, green represents wild birds, blue represents chickens, red represents ducks, dark green represents geese, light blue represents the environment, and pink represents human.

**Figure 7 viruses-15-01547-f007:**
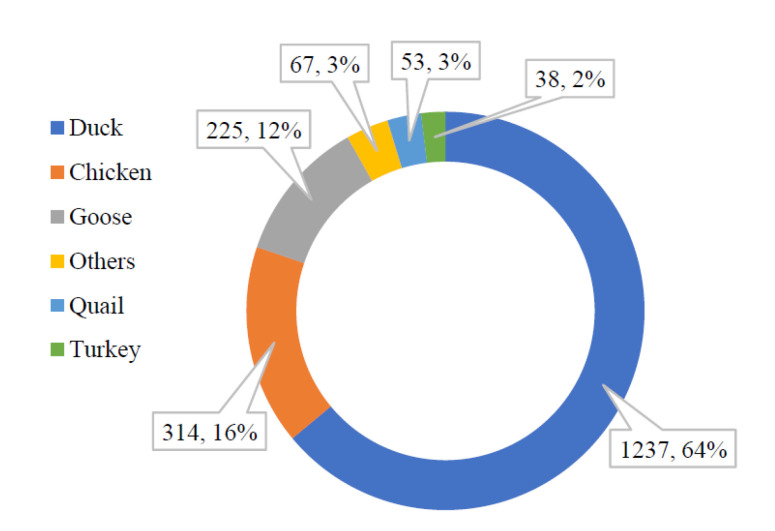
Distribution of H6 AIV in poultry. The numbers of H6 virus isolates in different domestic poultry are shown according to their submission information in GISAID. The number on the left represents its exact quantity, and the number on the right represents its percentage. For example, there are 1237 duck source sequences, which account for about 64% of the poultry source sequence, which is 1237.64%.

**Figure 8 viruses-15-01547-f008:**
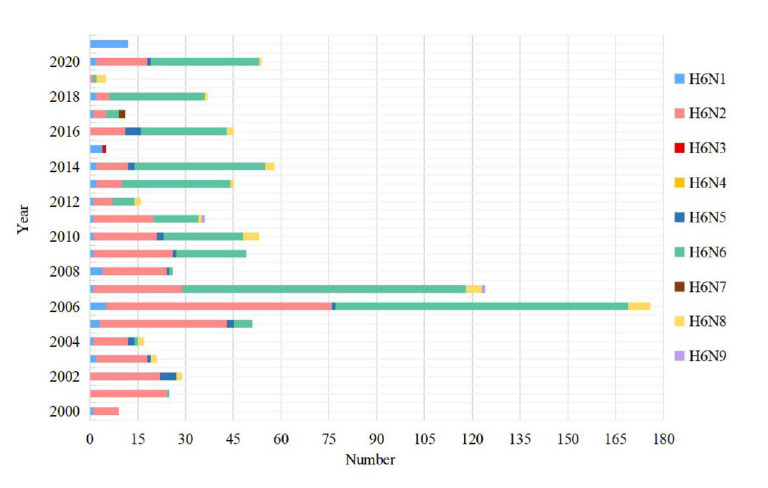
The number of H6 AIV subtypes of sequences isolated from ducks from 2000 to 2021.

**Figure 9 viruses-15-01547-f009:**
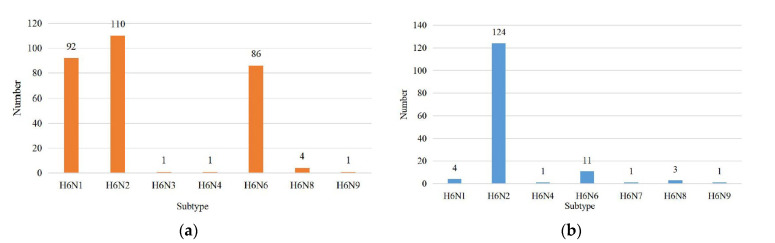
AIV H6 subtypes isolated from chickens and geese. The HA sequences were downloaded from the GISAID databases. (**a**) H6 avian influenza virus subtypes isolated from chickens. (**b**) H6 avian influenza virus subtypes isolated from geese.

**Figure 10 viruses-15-01547-f010:**
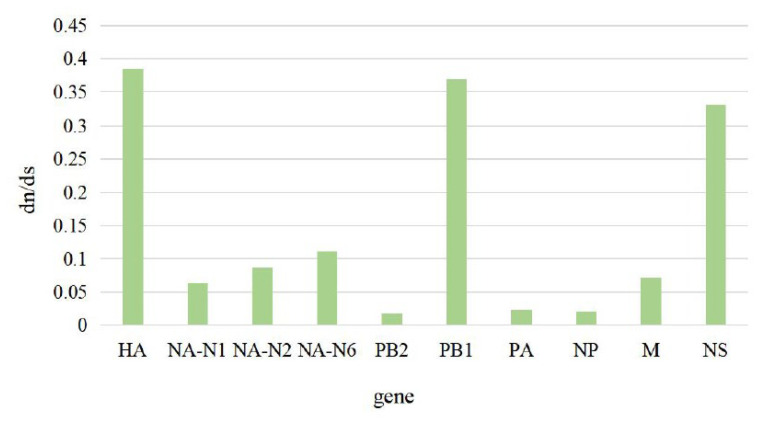
The result of selection Analysis. The associated amino acid changes were analyzed using MEGA 7.0. Consensus sequences were aligned, and mutations were recorded. The positions of the mutations for each enzootic cluster were confirmed manually. The number of amino acid changes in each enzootic cluster was counted.

**Figure 11 viruses-15-01547-f011:**
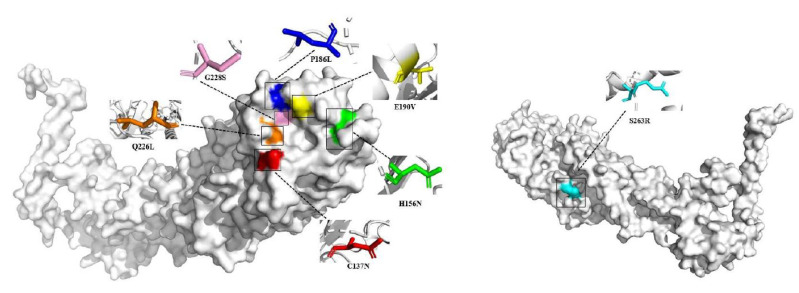
Structural simulation of the H6 avian influenza virus HA protein. In this figure, except for the E190V and C137N mutations, the other labeled amino acid mutations were artificially altered by PyMOL.

**Table 1 viruses-15-01547-t001:** The result of the DN/DS calculation. Rates of nonsynonymous and synonymous substitutions (dN/dS) for each segment from H6NX AIV were estimated using the Launch DnaSP6 software and MEGA 7.

Mean dn/ds	dn	ds	dn/ds
HA	0.09073	0.23544	0.385364
NA-N1	0.02619	0.41272	0.063457
NA-N2	0.0305	0.35173	0.086714
NA-N6	0.02182	0.19602	0.111315
PB2	0.00885	0.50059	0.017679
PB1	0.07201	0.19531	0.368696
PA	0.0093	0.40226	0.023119
NP	0.00731	0.36526	0.020013
M	0.01694	0.23705	0.071462
NS	0.10395	0.31309	0.332013

## Data Availability

No new data were created or analyzed in this study. Data sharing is not applicable to this article.

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
