# Peer review of "Evolution and Reassortment of H6 Subtype Avian Influenza Viruses"

_viruses, 2023, doi:10.3390/v15071547_

Round 1
Reviewer 1 Report
The review article by Lin et al., examines the evolution, prevalence and reassortment of H6 avian influenza A viruses in wild birds and poultry. They review the global prevalence of the viruses in nature, and the strains prevalent in China, as well as history of reassortments in the last 20 years. Further they look at its distribution in wild birds and poultry. Additionally, they examine the changes in the virus over time and map some of these on the structure of H6.
I have some comments for improvement of the manuscript.
The abstract does not really capture the subjects covered in the review but focuses on a perceived global threat.
Other reviews on this subject have been written (ref 5). This should be acknowledged and clearly state how this review differs.
In several places in figure legends there is mention of H6N0. What does this mean. No NA typing was done? Please define somewhere.
There are two Figure 2s in my version of the .pdf.
Figure 5 is ambiguous. I do not know what is being analyzed here. Do these sequence changes reflect drift and shift (reassortment). Both the writing in the text, and the figure legend are vague. Also Table 1 is missing.
The discussion could use some editing for clarity. It is not clear that this is actually a discussion following analysis done in the review, but presents more data analysis and new ideas. This could be put in review format with appropriate headings, such as distribution in wild birds, prevalence in poultry, and potential for vaccination and antivirals.
The discussion of vaccination comes up as the last idea. I am not really convinced that vaccination of poultry is necessary against this low pathogenic strain. It should be carefully considered that this could have the effect of driving pathogenicity of the virus.
Minor changes
Line 25. Highly pathogenic verses low pathogenic viruses are defined in chickens, so this should be stated.
Line 91, I think epidemic is not the correct word.
Line 121 and ref 19. I question this statement. I am not aware that viruses readily mix between European and North American flyways. I believe it is rather rare.
Phylogenetic analysis of viruses is not shown. It is difficult to follow the discussion without a figure.
Figure 10 is out of place where it is in the discussion. It could be more useful in the section discussing changes in the HA receptor.
Lines 154-58 are unclear. It is not clear how mutations in chickens indicate adaptation to mammals.
Line 171 does not make much sense as written. Is this referring to a co-infection study? Also, I think adaptors is the incorrect word. Perhaps adaptations?
Line 184 Mutational loci ( do you mean sites of mutations in the protein, or locus)
Line 199 adaptive fitness (not finesse).
Line 258 unclear meaning of this sentence.
Line 307 remove the word “list”. There is only one human infection.
Line 318 do you mean “impartial”
A few mistakes in word choice are noted in document.
Author Response
- The abstract does not really capture the subjects covered in the review but focuses on a perceived global threat.
Answer: Thank you for your suggestion. I had revised it. Thanks again.
- Other reviews on this subject have been written (ref 5). This should be acknowledged and clearly state how this review differs.
Answer: Thank you for your suggestion. The differences between the two articles are as follows:
- In this study, 3234 H6 AIV(HA) sequences were obtained from the GISAID and NCBI database as of October 3, 2022, which is more comprehensive in terms of quantity and year, and the statistical analysis results based on these sequences are expected to be more consistent with the current prevalence and evolution.
- This study focuses on the correlation between bird migration and global distribution of H6 AIV, mainly focusing on the reassortment between wild birds and poultry and between North American lineages and Eurasian lineages (mainly Asian countries), which provides reference data for subsequent studies of the transcontinental spread of wild bird source H6 AIV. In addition, this study plotted the number and subtype distribution of H6 AIV in China, highlighting the epidemic characteristics of this subtype in key distribution areas. More importantly, this study focuses on the analysis of the prevalence of H6 AIV in poultry (chicken, duck, and goose), which provides sufficient data information for clarifying the characteristics of H6 and NA subtypes prevalent in poultry and the focus of monitoring. In particular, this paper emphasizes China, a country with a high detection rate of the H6 subtype avian influenza virus.
- This study cites relatively new literature showing the amino acid mutation sites currently present in H6 AIV, such as M2-S31N and M2-V27I. It is supplemented with new functions such as Q226L(HA) and a combination of amino acid site mutations that may replace PB2-E627K, namely H156N, S263R(HA), and I38M (PA). This can provide a more systematic and comprehensive reflection of the current evolution of H6 AIV, suggesting that people should strengthen surveillance. This article supplements the latest research results of H6N2 and H6N6, such as finding that many H6N2 strains prefer human receptors, and some H6N6 viruses can cause severe clinical symptoms and even death in mice.
- We performed the selection analysis of H6 AIV, which helps to understand changes in adaptive trends to the host.
Thanks again.
- In several places in figure legends there is mention of H6N0. What does this mean. No NA typing was done? Please define somewhere.
Answer: Thank you for your suggestion. There were many H6N0 stains in GISAID and NCBI databases. No NA typing was done as you said indeed. I had stated it in the revised manuscript. Thanks again.
- There are two Figure 2s in my version of the .pdf.
Answer: Thank you for your suggestion. I am sorry to make a mistake. I had revised it in. revised manuscript. Thanks again.
- Figure 5 is ambiguous. I do not know what is being analyzed here. Do these sequence changes reflect drift and shift (reassortment). Both the writing in the text, and the figure legend are vague. Also Table 1 is missing.
Answer: Answer: Thank you for your suggestion. I had revised it in revised manuscript and upload the table 1. Thanks again.
- The discussion could use some editing for clarity. It is not clear that this is actually a discussion following analysis done in the review, but presents more data analysis and new ideas. This could be put in review format with appropriate headings, such as distribution in wild birds, prevalence in poultry, and potential for vaccination and antivirals.
Answer: Thank you for your suggestion. I had revised it. Thanks again.
- The discussion of vaccination comes up as the last idea. I am not really convinced that vaccination of poultry is necessary against this low pathogenic strain. It should be carefully considered that this could have the effect of driving pathogenicity of the virus.
Answer: Thank you for your suggestion. vaccination is the most efficient for disease. prevention and control. The strains of H6 subtype increased rapidly recently years. It is necessary to consider the stockpiling of vaccines such as H9N2 AIV. However, for H6 subtype, the surveillance is still the first and the most important task. Thanks again.
Minor changes
- Line 25. Highly pathogenic verses low pathogenic viruses are defined in chickens, so this should be stated.
Answer: Thank you for your suggestion. Highly pathogenic verses low pathogenic viruses are defined in chickens. I had revised it. Thanks again.
- Line 91, I think epidemic is not the correct word.
Answer: Thank you for your suggestion. The word “epidemic” in here was not suitable. I had revised it. Thanks again.
- Line 121 and ref 19. I question this statement. I am not aware that viruses readily mix between European and North American flyways. I believe it is rather rare.
Answer: Thank you for your suggestion. The paper (ref 19) indicates that the PB1 gene in the sequence (A/ML/AH/1-451/2019(H6N2) belongs to the North American lineage, while the other internal genes belong to the Eurasian lineage. The Eurasian lineage referred to here does not indeed refer to European countries, but to the Asian countries of Japan, Korea and Mongolia. Thanks again.
- Phylogenetic analysis of viruses is not shown. It is difficult to follow the discussion without a figure.
Answer: Thank you for your suggestion. I had supplied the phylogenetic tree in the revised article. Thanks again.
- Figure 10 is out of place where it is in the discussion. It could be more useful in the section discussing changes in the HA receptor.
Answer: Thank for your suggestion. I had revised it.
- Lines 154-58 are unclear. It is not clear how mutations in chickens indicate adaptation to mammals.
Answer: Thank you for your suggestion. My previous description was too categorical, and I had modified it to use words like “may”. The mechanism of transmission of avian influenza has yet to be studied. Thanks again.
- Line 171 does not make much sense as written. Is this referring to a co-infection study? Also, I think adaptors is the incorrect word. Perhaps adaptations?
Answer: Thank you for your suggestion. In the study, authors obtained two plaque-purified mouse-adapted variants of the original SJ/275 virus for further characterization and named MA-P8M1 and MA-P8M3. Then, they infected the mice with WT H6N1 SJ/275, MA-P8M1, and MA-P8M3, respectively. I had revise it. Thanks again.
- Line 184 Mutational loci ( do you mean sites of mutations in the protein, or locus)
Answer: Thank you for your suggestion. I mean sites of mutations in protein. I had revised it. Thanks again.
- Line 199 adaptive fitness (not finesse).
Answer: Thank you for your suggestion. I had revised it. Thanks again.
- Line 258 unclear meaning of this sentence.
Answer: Thank you for your suggestion. This sentence refers to the high H6 AIV positive detection rate of chickens and geese in poultry. According to the statistical results (sequences from GISIAD database), they account for 16% and 12% of poultry, respectively. Thanks again.
- Line 307 remove the word “list”. There is only one human infection.
Answer: Thank you for your suggestion. I had revised it.
- Line 318 do you mean “impartial”
Answer: Thank you for your suggestion. I had revised it. Thanks again.
- A few mistakes in word choice are noted in document.
Answer: Thank you for your suggestion. I had made changes to some of the words that are not appropriate. Thanks again.
Reviewer 2 Report
The H6 subtype of avian influenza virus (H6 AIV) causes economic losses to the poultry industry and threaten to human health. This study focuses on the prevalence of H6 AIV in poultry and wild birds, genetic variation characteristics of the H6NX AIV, for the scientific prevention and control of H6 AIV.
There were some questions as follow:
1. There were some grammatical mistakes, you should find for a native English person to polishing up you’re writing.
2. Some sentences do not suggest appearing in the manuscript, such as line 62-64. It is not suitable for review article.
3. In review, there was so much in discussions, it would better to put this part inserted in the previous description.
There were some grammatical mistakes, you should find for a native English person to polishing up you’re writing.
Author Response
Review 2
There were some questions as follow:
- There were some grammatical mistakes, you should find for a native English person to polishing up you’re writing.
Answer: Thank you for your suggestion. I had revised the manuscript by AJE company before submitted. I had polished up my writing again. Thanks again.
- Some sentences do not suggest appearing in the manuscript, such as line 62-64. It is not suitable for review article.
Answer: Thank you for your suggestion. I had revised that in revised manuscript. Thanks again.
- In review, there was so much in discussions, it would better to put this part inserted in the previous description.
Answer: Thank you for your suggestion. I had deleted the discussion part and insert into the proper place in revised manuscript. Thanks again.